# Positive Touch Deprivation during the COVID-19 Pandemic: Effects on Anxiety, Stress, and Depression among Italian General Population

**DOI:** 10.3390/brainsci13040540

**Published:** 2023-03-24

**Authors:** Francesco Bruno, Carlotta Tagliaferro, Sonia Canterini, Valentina Laganà, Marianna Contrada, Chiara Fioravanti, Natalia Altomari, Rebecca Pistininzi, Francesca Tarantino, Alessia Placanica, Ersilia Martina Greco, Francesca Capicotto, Saverio Spadea, Anna Maria Coscarella, Martina Bonanno, Federica Scarfone, Sara Luchetta-Mattace, Alessia Filice, Annamaria Pettinato, Ana Avramovic, Chloe Lau, Georgia Marunic, Francesca Chiesi

**Affiliations:** 1Department of Primary Care, Regional Neurogenetic Centre (CRN), ASP CZ, 88046 Lamezia Terme, Italy; 2Association for Neurogenetic Research (ARN), 88046 Lamezia Terme, Italy; 3Academy of Cognitive Behavioral Sciences of Calabria (ASCoC), 88046 Lamezia Terme, Italy; 4School of Psychology, University of Florence, 50135 Florence, Italy; 5Division of Neuroscience, Department of Psychology, University La Sapienza, 00158 Rome, Italy; 6S. Anna Institute and Research in Advanced Neurorehabilitation (RAN), 88900 Crotone, Italy; 7Psychiatrie und Psychotherapie -Kinder- und Jugendpsychiatrie, Zürcherstrasse, 8852 Altendorf, Switzerland; 8Department of Mathematics and Computer Science, University of Calabria, 87036 Rende, Italy; 9Scholar at Department of Medical and Surgical Sciences, Degree Course in Science and Techniques of Cognitive Psychology, Magna Graecia University of Catanzaro, 88100 Catanzaro, Italy; 10Faculty of Philosophy, Laboratory for Experimental Psychology, University of Belgrade, 11102 Belgrade, Serbia; 11Department of Psychology, University of Western Ontario, London ON N6A 3K7, Canada; 12Department of Neuroscience, Psychology, Drug, and Child’s Health (NEUROFARBA), Section of Psychology, University of Florence, 50135 Florence, Italy

**Keywords:** positive touch, touch deprivation, anxiety, stress, depression, hugs, kisses, caresses, hold hands, COVID-19 Pandemic

## Abstract

Physical distancing due to the COVID-19 Pandemic has limited the opportunities for family members, friends, and significant others to show physical affection (i.e., hugs, kisses, caresses, holding hands) during social interactions. The present study investigated the effects of positive touch and psychological distress in 991 Italian participants (*M*_age_ = 34.43, *SD* = 14.27). Results showed the frequency of hugs with the cohabiting partner significantly decreased the symptoms of depression (*β* = −1.187, *p* = 0.018, *e^β^ =* 0.30, 95% CI = 0.11–0.82), whereas the frequency of caresses with cohabiting relatives predicted the symptoms of anxiety (*β* = 0.575, *p* = 0.034, *e^β^* = 1.78, 95% CI = 1.04–3.03). The frequency of hugs (*β* = −0.609, *p* = 0.049, *e^β^ =* 0.54, 95% CI = 0.30–1.00), and kisses (*β* = 0.663, *p* = 0.045, *e^β^* = 1.94, 95% CI = 1.01–3.71) with non-cohabiting relatives predicted the symptoms of anxiety (*χ*^2^ = 1.35, *df* = 5, *p* = 0.93). These results suggest the importance of positive touch on psychological well-being in the social context.

## 1. Introduction

The Coronavirus Disease 2019 (COVID-19) Pandemic, caused by the SARS-CoV-2 virus was first reported in December 2019 in Wuhan China and subsequently documented in more than 210 countries around the world, represented an emergency for international public health [1] of both the general population [2,3,4] and vulnerable groups [5,6,7].

In many countries, various measures have been adopted in attempt to contain the spread of this infection, including social or physical distancing. The goals of physical distancing were to decrease the contacts of infected people with uninfected, minimize the virus’s transmission, and decrease associated morbidity and mortality [8].

Physical distancing has limited the opportunities for family members, friends, and colleagues to show physical affection during social interactions. Positive touch (i.e., hugs, kisses, caresses, holding hands) represents a fundamental element of the human experience, as it constitutes a primary component of socio-emotional, cognitive, physical, and neurological development [9,10]. Several studies indicate that this important form of non-verbal communication allows humans to express and transmit social support and a wide range of primary (i.e., happiness and sadness) and pro-social (i.e., love, gratitude, sympathy) emotions [10]. Moreover, positive touch is associated with the activation of the neuroendocrine system with the consequent release of oxytocin [11,12]. In turn, oxytocin may influence the levels of expression of dopamine, serotonin, and endorphins, all of which are associated with positive emotions [10]. On a psychological level, these effects result in the reduction of interpersonal conflicts, increased attachment and bonding between people, improved recognition of emotions and social memory, and greater levels of empathy, sincerity, and fidelity [10,11,12]. Furthermore, several studies have suggested that the social support and emotions transmitted through positive touch are able to mitigate the stress response by reducing cortisol levels. In turn, lowered levels of perceived stress have a protective effect on both cardiovascular and immune systems [10,11,12,13].

Although social distancing represents an indispensable measure to contain the spread of the COVID-19 Pandemic, it limits the individual’s ability to maintain emotional and social connections and thus could have dramatic consequences on mental health. Public health regulations of the social distancing due to the COVID-19 Pandemic was associated with a higher desire for touch, which in turn was related to an increased perceived pleasantness of observing touch [14]. Moreover, the deprivation of intimate touch (e.g., kiss, hugs, caress from partner or close family) due to the COVID-19-related restrictions was associated with anxiety and loneliness [15]. Its associations between psychological distress and both personal distancing behavior [16] and touch deprivation [17] were observed during the U.S. COVID-19 lockdown. Conversely, the frequency of both in-person social and sexual connections were generally associated with lower levels of depressive symptoms [18].

To our knowledge, no study has separately examined the frequency reduction of each type of positive touch (i.e., kisses, hugs, caresses, holding hands) depending on the type of relationship with cohabitants and non-cohabiting people as well as the weight that this could have on psychological distress during the COVID-19 Pandemic.

Thus, the aims of this study were to: (i) estimate the frequency of the reduction of positive touch (i.e., hugs, kisses, caresses, holding hands) both with cohabiting (i.e., partners, children, relatives or friends/roommates) and non-cohabiting (i.e., partners, children, relatives and friends) people due to the COVID-19 Pandemic; (ii) understand the impact of the frequency reduction of positive touch both with cohabiting and non-cohabiting people on psychological distress (i.e., symptoms of depression, anxiety, and stress).

## 2. Materials and Methods

### 2.1. Participants and Procedure

The questionnaire, in electronic format, was proposed to a convenience sample and distributed widely over a period of COVID-19 restrictions between May and June 2021. Sampling was based on the ‘snowball’ method [19], in which non-graduate students of a psychology course were invited to participate in a study with an online questionnaire and were encouraged to recruit their acquaintances and relatives as well; in addition, the questionnaire was also shared on social network groups to increase sample diversity. In all, 3003 questionnaires were collected from the general Italian population. From these, approximately one third were randomly drawn to constitute the sample for this study, more precisely, 991 Italian participants (57% female, age range from 18 to 85 years, *M_age_* = 34.43, *SD* = 14.27).

Most participants have completed high school (52.7%), lived with their family members (41.1%), and were engaged to a romantic partner (37.6%). All participants were aware that participation in the study was voluntary, and data were collected anonymously for research purposes. All participants consented to participate under these terms and debriefed upon completion of the study. No compensation or incentives were provided. Approval for this study was obtained from the Ethical Committee of Calabria Region (Catanzaro, Italy; Prot. n. 52098, 16 April 2021).

### 2.2. Measures

The frequency of four physical positive touch exposures (i.e., hugging, kissing, caressing, holding hands) with reference to the period before the COVID-19 Pandemic and during the last month was assessed. Each response category had four options: not at all, once or a few times, 1–3 times a week, and almost every day. Social connections were measured by asking participants to refer both to touch with cohabiting persons (partners/children/relatives/friends) and to positive touch with non-cohabiting persons (children/relatives/friends). Each type of physical positive touch was correlated with measures of symptoms of depression, anxiety, and stress by administering the Depression Anxiety Stress Scales-21 (DASS-21) [20], Italian version [21]. This scale is a self-report questionnaire with 21-items measuring symptoms of depression, stress, and anxiety (seven items for each subscale) based on a four-point rating scale (with anchors labeled 0 = did not apply to me at all and 3 = applied to me much, or most of the time). A high score on each subscale indicates elevated symptoms of depression, anxiety, or stress. In the current sample, Cronbach’s Alpha for Stress and Depression subscales were excellent (α = 0.93 and α = 0.92, respectively) and good for the Anxiety subscale (α = 0.88).

### 2.3. Statistical Analysis

To account for any differential nonresponse, low amounts of missing data (<5% for all variables) were considered a complete case. We calculated descriptive estimates of the prevalence of the four physical touch before the pandemic and during the last month for each condition of cohabitation or non-cohabitation. To compare the exposures during the two timeframes, the Wilcoxon test for correlated samples was used. Log-binomial regression with adjusted odds ratios and 95% CIs were computed to estimate the associations between each of the four exposures and the three mental health outcomes (depression, anxiety, and stress). To calculate the regression outcomes, depression, anxiety, and stress were transformed into binary variables. The scale cutoff of greater than or equal to 7 was used to identify those likely experiencing significant depressive symptoms. A scale cutoff of 6 and 10 was used to identify significant anxiety and stress symptoms, respectively. In the binomial logistic regression, the Hosmer and Lemeshow tests were used to evaluate the goodness of fit of the models (*p* > 0.05) [22]. All analyses were conducted using SPSS 24.0 statistical software.

## 3. Results

The valid percentage of moderate/extremely severe depression, anxiety, and stress symptoms reported in the past seven days were 37.4%, 24.9%, and 33.3%, respectively. In Table 1, the frequencies of physical positive touch with cohabiting people are reported. Persons living with partners (*N* = 561), children (*N* = 248), relatives (*N* = 453), and friends (*N* = 61) seem to report a statistically significant reduction in physical touch between before the pandemic and during the last month according to the Wilcoxon test (*p* < 0.001). The frequencies of physical positive touch with non-cohabiting people are presented in Table 2. Similarly, people not living with their children (*N* = 87) and relatives and friends (*N* = 991) seem to report a statistically significant reduction in physical touch between before the pandemic and in the last month according to the Wilcoxon test (*p* < 0.001). From these findings, the adequacy of the models was checked by using as predictors the types of positive physical contact and as outcomes the levels of self-assessed depressive symptoms for both cohabiting (i.e., partners, children, relatives, or friends/roommates) and non-cohabiting (i.e., children, relatives, and friends) people. Different models were evaluated and all were found to be non-significant, allowing the data within them to be interpreted. However, only certain types of positive physical contact were found to predict significant levels of self-assessed depressive, anxiety, and stress symptoms during the pandemic. All types of physical positive touch were not significant (*p* > 0.05) predictors of self-assessed anxiety, stress, and depression symptoms for cohabiting and non-cohabiting children and for cohabiting and non-cohabiting friends (full data are available in the Appendix A). In the current study, the model that associates the frequency of hugs with the partner and the decreased levels of depression (*β* = −1.187, *p* = 0.018, *e^β^* = 0.30, 95% CI = 0.11–0.82) showed the Hosmer and Lemeshow test (*χ*^2^ = 3.13, *df* = 4, *p* = 0.54) was not significant. This indicated good fit with a significance level greater than 0.05, and the estimated variance explained by Nagelkerke *R*^2^ of 0.05. Similarly, the frequency of caresses with cohabiting relatives predicted the levels of anxiety (*β* = 0.575, *p* = 0.034, *e^β^ =* 1.78, 95% CI = 1.04–3.03) and the Hosmer–Lemeshow test result confirmed that the model had a good fit (*χ*^2^ = 1.67, *df* = 6, *p* = 0.95), with a Nagelkerke *R*^2^ of 0.02. The frequency of hugs (*β*= −0.609, *p* = 0.049, *e^β^* = 0.54, 95% CI = 0.30–1.00) and kisses (*β*= 0.663, *p* = 0.045, *e^β^* = 1.94, 95% CI= 1.01–3.71) with non-cohabiting relatives predicted anxiety level (*χ*^2^
*=* 1.35, *df* = 5, *p* = 0.93), with a Nagelkerke *R*^2^ value of 0.01 (Table 3).

## 4. Discussion

The general aim of this study was to estimate the frequency of the reduction of positive touch (i.e., hugs, kisses, caresses, holding hands) both with cohabiting (i.e., partners, children, relatives, or friends/roommates) and non-cohabiting (i.e., children, relatives, and friends) people due to the COVID-19 Pandemic in the Italian general population. In addition, we were interested in the impact of the frequency reduction of positive touch both with cohabiting and non-cohabiting people on psychological distress (i.e., symptoms of depression, anxiety, and stress).

As expected, compared to before the pandemic, we found a significant and general reduction in the frequency of positive touch (hugs, kisses, caresses, holding hands) both with cohabiting (i.e., partners, children, relatives, or friends/roommates) and non-cohabiting (i.e., partners, children, relatives, and friends) people. These results are consistent with the social distancing measures provided by the Italian Ministry of Health, such as maintaining a distance of at least 1 m from other people and wearing the ffp2 mask in open and closed places to limit the spread of COVID-19 [23]. In line with our findings, other studies reported a general reduction of positive touch (i.e., hugs, kisses, and caresses) from partners (von Mohr et al., 2021; Field et al., 2020), children (Field et al., 2020), close family members (von Mohr et al., 2021), friends, and work colleagues or carers [15] during the first wave of the COVID-19 Pandemic and due to COVID-19-related restrictions.

Moreover, we reported that the frequency of hugs with the cohabiting partner significantly decreased the symptoms of depression. This finding is in line with the results of Field et al. (2020) who reported that the frequency of touch with partners was related to lower levels of depressive symptoms during the U.S. COVID-19 lockdown. This effect is likely attributable to the well-documented increase in oxytocin levels that occurs when more hugs are received from the partner [10,11]. In addition, another line of research suggests that high and low levels of oxytocin are associated with high and low levels of depressive symptoms, respectively (for a review see: McQuaid et al., 2014). On the other hand, we found that the frequency of caresses with cohabiting relatives and the frequency of hugs and kisses with non-cohabiting relatives predicted symptoms of anxiety (Figure 1).

These effects are likely due to the fact that, compared to the partner, there is less trust and knowledge of the social networks frequented and of the physical contacts exchanged by co-habitants and non-cohabitant relatives. As a consequence, having positive physical contacts with them could increase the fear of being infected with SARS-CoV-2, given that both kisses and caresses are means of contracting it. However, other studies are needed to better investigate these aspects.

There are considerable limitations to this research that can be helpful for future studies. First, it is important to recognize that sampling used is not as effective as true random sampling; nonetheless, it allowed us to overcome specific disadvantages connected with true random sampling, such as being overly expensive and time-consuming. Second, self-reported measures were administered to assess the dimensions of this study. Future research should take into consideration different methods (e.g., clinician-ratings, peer-ratings) to reduce the influence of self-report bias. Finally, an individual’s retrospective report of physical touch prior to the COVID-19 Pandemic is dependent on their memory. Future studies should employ diary studies to address this limitation.

## 5. Conclusions

In conclusion, in this study, we reported a general reduction in the frequency of positive touch (i.e., hugs, kisses, caresses, holding hands) between before the COVID-19 Pandemic and in the last month, both with cohabiting and non-cohabiting people, in line with the social distancing measures adopted by the Italian government. Interestingly, the frequency of hugs with the cohabiting partner significantly decreased the symptoms of depression, whereas the frequency of caresses with cohabiting relatives and the frequency of hugs and kisses with non-cohabiting relatives predicted anxiety levels. This may be due to social networks and frequency of physical contacts exchanged by the relative are less known than those of the cohabiting partner. Further studies are needed to better investigate these aspects.

## Figures and Tables

**Figure 1 brainsci-13-00540-f001:**
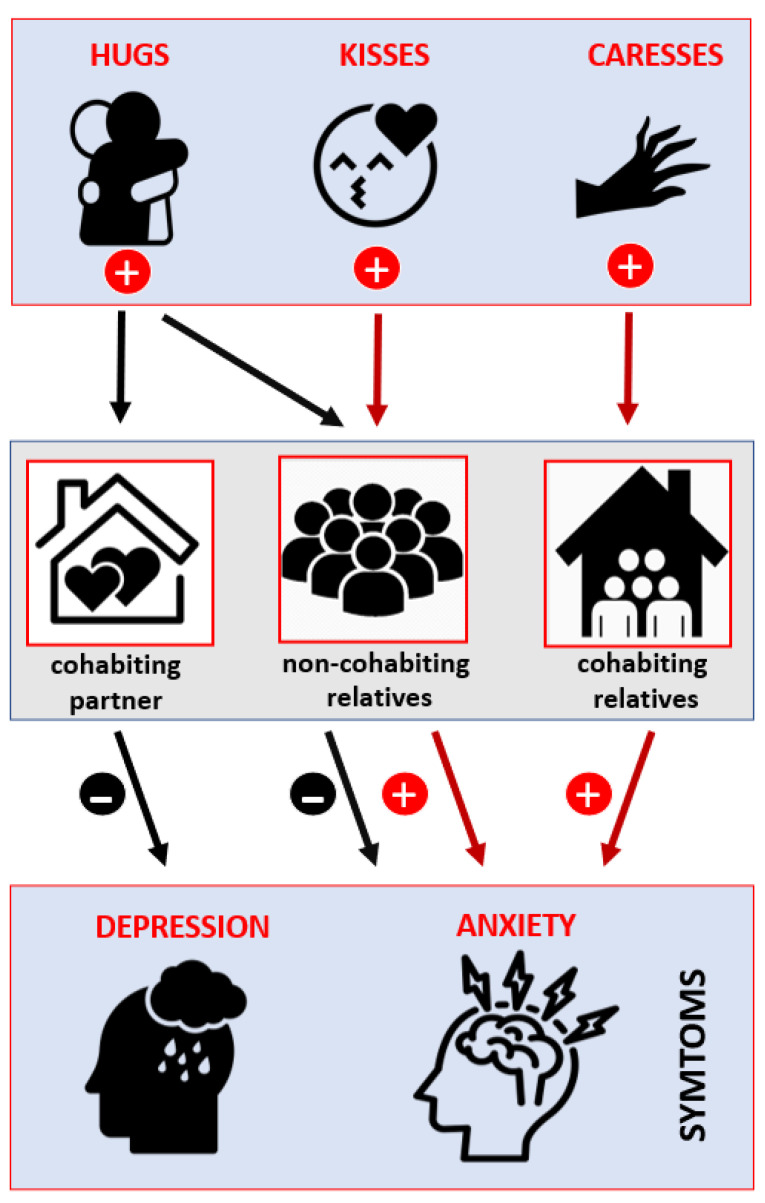
Effects of positive touch on psychological distress during the COVID-19 Pandemic.

**Table 1 brainsci-13-00540-t001:** Frequency of reporting physical positive touch before pandemic and in the last month with cohabiting people.

		Before Pandemic *N* (%)	In the Last Month *N* (%)	*p*-Value
How often have you hugged your partner?	Almost everyday	376 (67.0)	309 (55.1)	<0.001
	1–3 times a week	150 (26.7)	169 (30.1)	
	Once or a few times	31 (5.5)	69 (12.3)	
	Not at all	4 (0.7)	14 (2.5)	
How often have you kissed your partner?	Almost everyday	373 (66.5)	305 (54.4)	<0.001
	1–3 times a week	150 (26.7)	169 (30.1)	
	Once or a few times	33 (5.9)	70 (12.5)	
	Not at all	5 (0.9)	17 (3.0)	
How often have you caressed your partner?	Almost everyday	373 (66.5)	309 (55.1)	<0.001
	1–3 times a week	148 (26.4)	163 (29.1)	
	Once or a few times	35 (6.2)	73 (13.0)	
	Not at all	5 (0.9)	16 (2.9)	
How often have you held your partner’s hands?	Almost everyday	364 (64.9)	302 (53.8)	<0.001
	1–3 times a week	153 (27.3)	159 (28.3)	
	Once or a few times	40 (7.1)	83 (14.8)	
	Not at all	4 (0.7)	17 (3.0)	
How often have you hugged your children?	Almost everyday	203 (81.9)	185 (74.6)	<0.001
	1–3 times a week	29 (11.7)	31 (12.5)	
	Once or a few times	14 (5.6)	31 (12.5)	
	Not at all	2 (0.8)	1 (0.4)	
How often have you kissed your children?	Almost everyday	192 (77.4)	179 (72.2)	0.002
	1–3 times a week	33 (13.3)	32 (12.9)	
	Once or a few times	19 (7.7)	33 (13.3)	
	Not at all	4 (1.6)	4 (1.6)	
How often have you caressed your children?	Almost everyday	198 (79.8)	180 (72.6)	0.001
	1–3 times a week	27 (10.9)	33 (13.3)	
	Once or a few times	20 (8.1)	32 (12.9)	
	Not at all	3 (1.2)	3 (1.2)	
How often have you held your children’s hands?	Almost everyday	193 (77.8)	177 (71.4)	0.001
	1–3 times a week	33 (13.8)	34 (13.7)	
	Once or a few times	19 (7.7)	34 (13.7)	
	Not at all	3 (1.2)	3 (1.2)	
How often have you hugged your relatives?	Almost everyday	217 (47.9)	145 (32.0)	<0.001
	1–3 times a week	95 (21.0)	107 (23.6)	
	Once or a few times	123 (27.2)	158 (34.9)	
	Not at all	18 (4.0)	43 (9.5)	
How often have you kissed your relatives?	Almost everyday	198 (43.7)	127 (28.0)	<0.001
	1–3 times a week	88 (19.4)	87 (19.2)	
	Once or a few times	138 (30.5)	177 (39.1)	
	Not at all	29 (6.4)	62 (13.7)	
How often have you caressed your relatives?	Almost everyday	196 (43.3)	132 (29.1)	<0.001
	1–3 times a week	90 (19.9)	88 (19.4)	
	Once or a few times	128 (28.3)	171 (37.7)	
	Not at all	39 (8.6)	62 (13.7)	
How often have you held your relatives’ hands?	Almost everyday	203 (44.8)	143 (31.6)	<0.001
	1–3 times a week	87 (19.2)	81 (17.9)	
	Once or a few times	128 (28.3)	168 (37.1)	
	Not at all	35 (7.7)	61 (13.5)	
How often have you hugged your friends?	Almost everyday	19 (31.1)	11 (18.0)	<0.001
	1–3 times a week	18 (29.5)	15 (24.6)	
	Once or a few times	17 (27.9)	21 (34.4)	
	Not at all	7 (11.5)	14 (23.0)	
How often have you kissed your friends?	Almost everyday	18 (29.5)	9 (14.8)	<0.001
	1–3 times a week	10 (16.4)	12 (19.7)	
	Once or a few times	21 (34.4)	22 (36.1)	
	Not at all	12 (19.7)	18 (29.5)	
How often have you caressed your friends?	Almost everyday	19 (31.1)	10 (16.4)	<0.001
	1–3 times a week	9 (14.8)	12 (19.7)	
	Once or a few times	19 (31.1)	20 (32.8)	
	Not at all	14 (23.0)	19 (31.1)	
How often have you held your friends’ hands?	Almost everyday	20 (32.8)	11 (18.0)	<0.001
	1–3 times a week	14 (23.0)	11 (18.0)	
	Once or a few times	19 (31.1)	23 (37.7)	
	Not at all	8 (13.1)	16 (26.2)	

Notes. *N* = frequency, % = valid percentage, *p*-value = Wilcoxon test for correlated samples.

**Table 2 brainsci-13-00540-t002:** Frequency of reporting physical positive touch before the COVID-19 Pandemic and in the last month with non-cohabiting people.

		Before Pandemic *N* (%)	In the Last Month *N* (%)	*p*-Value
How often have you hugged your children?	Almost everyday	13 (14.9)	6 (6.9)	<0.001
	1–3 times a week	38 (43.7)	25 (28.7)	
	Once or a few times	36 (41.4)	47 (54.0)	
	Not at all	0 (0.0)	9 (10.3)	
How often have you kissed your children?	Almost everyday	12 (13.8)	6 (6.9)	<0.001
	1–3 times a week	36 (41.4)	23 (26.4)	
	Once or a few times	37 (42.5)	47 (54.0)	
	Not at all	2 (2.3)	11 (12.6)	
How often have you caressed your children?	Almost everyday	12 (13.8)	6 (6.9)	<0.001
	1–3 times a week	37 (42.5)	23 (26.4)	
	Once or a few times	37 (42.5)	47 (54.0)	
	Not at all	1 (1.1)	11 (12.6)	
How often have you held your children’s hands?	Almost everyday	14 (16.1)	6 (6.9)	<0.001
	1–3 times a week	36 (41.4)	24 (27.6)	
	Once or a few times	36 (41.4)	46 (52.9)	
	Not at all	1 (1.1)	11 (12.6)	
How often have you hugged your relatives?	Almost everyday	102 (10.3)	40 (4.0)	<0.001
	1–3 times a week	374 (37.7)	131 (13.2)	
	Once or a few times	446 (45.0)	553 (55.8)	
	Not at all	69 (7.0)	267 (26.9)	
How often have you kissed your relatives?	Almost everyday	95 (9.6)	41 (4.1)	<0.001
	1–3 times a week	331 (33.4)	116 (11.7)	
	Once or a few times	474 (47.8)	530 (53.5)	
	Not at all	91 (9.2)	304 (30.7)	
How often have you caressed your relatives?	Almost everyday	93 (9.4)	40 (4.0)	<0.001
	1–3 times a week	294 (29.7)	115 (11.6)	
	Once or a few times	479 (48.3)	524 (52.9)	
	Not at all	125 (12.6)	312 (31.5)	
How often have you held your relatives’ hands?	Almost everyday	104 (10.5)	41 (4.1)	<0.001
	1–3 times a week	314 (31.7)	129 (13.0)	
	Once or a few times	479 (48.3)	527 (53.2)	
	Not at all	94 (9.5)	294 (29.7)	
How often have you hugged your friends?	Almost everyday	122 (12.3)	24 (2.4)	<0.001
	1–3 times a week	500 (50.5)	117 (11.8)	
	Once or a few times	319 (32.2)	555 (56.0)	
	Not at all	50 (5.0)	295 (29.8)	
How often have you kissed your friends?	Almost everyday	103 (10.4)	18 (1.8)	<0.001
	1–3 times a week	407 (41.1)	92 (9.3)	
	Once or a few times	370 (37.3)	495 (49.9)	
	Not at all	111 (11.2)	386 (39.0)	
How often have you caressed your friends?	Almost everyday	96 (9.7)	20 (2.0)	<0.001
	1–3 times a week	336 (33.9)	85 (8.6)	
	Once or a few times	396 (40.0)	506 (51.1)	
	Not at all	163 (16.4)	380 (38.3)	
How often have you held your friends’ hands?	Almost everyday	132 (13.3)	22 (2.2)	<0.001
	1–3 times a week	398 (40.2)	104 (10.5)	
	Once or a few times	359 (36.2)	522 (52.7)	
	Not at all	102 (10.3)	343 (34.6)	

Notes. *N* = frequency, % = valid percentage, *p*-value = Wilcoxon test for correlated samples.

**Table 3 brainsci-13-00540-t003:** Log binomial regression between the prevalence of the four physical positive touch types and each of the three mental health outcomes (depression, anxiety, and stress).

**DEPRESSION**							
**Cohabiting Partner**	** *β* **	** *SE β* **	**Wald’s χ^2^**	** *df* **	** *p* **	**Odds Ratio** **(*e^β^*)**	**95%CI** **(*e^β^*)**
How often have you hugged your partner?	−1.187	0.50	5.57	1	0.018	0.30	0.11–0.82
How often have you kissed your partner?	0.481	0.41	1.36	1	0.243	1.62	0.72–3.63
How often have you caressed your partner?	0.610	0.49	1.57	1	0.210	1.84	0.71–4.78
How often have you held your partner’s hands?	−0.332	0.34	0.96	1	0.327	0.72	0.37–1.39
Overall model evaluation: Goodness-of-fit test: Hosmer and Lemeshow: χ^2^ = 3.13, df = 4, *p* = 0.54. Nagelkerke *R*^2^ = 0.05.							
**ANXIETY**							
**Cohabiting Relatives**	** *β* **	** *SE β* **	**Wald’s χ^2^**	** *df* **	** *p* **	**Odds Ratio** **(*e^β^*)**	**95%CI** **(*e^β^*)**
How often have you hugged your relatives?	−0.304	0.25	1.49	1	0.223	0.74	0.45–1.20
How often have you kissed your relatives?	−0.31	0.23	1.77	1	0.184	0.73	0.46–1.16
How often have you caressed your relatives?	0.575	0.27	4.48	1	0.034	1.78	1.04–3.03
How often have you held your relatives’ hands?	−0.043	0.22	0.04	1	0.845	0.96	0.62–1.47
Overall model evaluation: Goodness-of-fit test: Hosmer and Lemeshow: χ^2^ = 1.67, df = 6, *p* = 0.95. Nagelkerke *R*^2^ = 0.02.							
**ANXIETY**							
**Non-Cohabiting Relatives**	** *β* **	** *SE β* **	**Wald’s χ^2^**	** *df* **	** *p* **	**Odds Ratio** **(*e^β^*)**	**95%CI** **(*e^β^*)**
How often have you hugged your relatives?	−0.609	0.31	3.86	1	0.049	0.54	0.30–1.00
How often have you kissed your relatives?	0.663	0.33	4.01	1	0.045	1.94	1.01–3.71
How often have you caressed your children?	−0.33	0.34	0.91	1	0.339	0.72	0.37–1.41
How often have you held your relatives’ hands?	0.176	0.27	0.42	1	0.519	1.19	0.70–2.03
Overall model evaluation: Goodness-of-fit test: Hosmer and Lemeshow: χ^2^ = 1.35, df = 5, *p* = 0.93. Nagelkerke *R*^2^ = 0.01.							

## Data Availability

The datasets generated during and/or analyzed during the current study are available from the corresponding author on reasonable request.

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
