# Peer review of "Positive Touch Deprivation during the COVID-19 Pandemic: Effects on Anxiety, Stress, and Depression among Italian General Population"

_brainsci, 2023, doi:10.3390/brainsci13040540_

Round 1

Reviewer 1 Report

Dear author, thank you on performing an interesting study on the very important topic of the importance of physical touch on psychological wellbeing. 

While the topic is relevant, study design is insufficient. You have appropriately used validated questionnaire but sampling was not ideal which you clearly mentioned in the limitation section.

More troubling is the statistical analysis. If done properly, most likely no significant finding would be reported. First of all, on one hand you are describing mean and standard deviation while on the other hand performing non parametric test such as Wilcoxon. Have you even considered data distribution?

All your findings related to regression models are based on significant variables inside the non-significant models. Additionally, some form of post-hoc analysis was performed such as Bonferroni correction should be considered.

The tables are filled with non-essential data and therefore very confusing. They should be merged and concentrated with only the most relevant data. Describing all sorts of statistical data could be described as a supplement. 

I believe the professional statistician should re-perform the analysis.

Author Response

Reviewer #1:

Comment 1. While the topic is relevant, study design is insufficient. You have appropriately used validated questionnaire but sampling was not ideal which you clearly mentioned in the limitation section.

Response. Thank you for this comment. The sampling method was already mentioned in the limitations of the study. We've now marked it in red so you can see it better.

Comment 2. More troubling is the statistical analysis. If done properly, most likely no significant finding would be reported. First of all, on one hand you are describing mean and standard deviation while on the other hand performing non parametric test such as Wilcoxon. Have you even considered data distribution?

Response. We thank the reviewer for the comment. We checked along the manuscript and did not find a reference to means and standard deviations. In fact, in Table 1 and Table 2, we report the frequencies of the different types of positive physical contact and their respective percentages. It is an accepted way of reporting this type of data that can be found in other studies that have adopted the same research methodology (Sammarra et al., 2022). Furthermore, we thank the reviewer for the suggestion to consider the distribution of the data. These have been checked and have a slight deviation from normality. A non-parametric test was chosen to perform on our data since, the Wilcoxon test is often used in situations where the underlying assumption of classifiability of differences between scale values is not fulfilled, such as in our study where the frequencies of physical contacts can be considered as ordinal variables (Oyeka, & Ebuh, 2012).

Source: https://doi.org/10.1016/j.yebeh.2022.108600

               https://doi.org/10.4236/ojs.2012.22019

Comment 3. All your findings related to regression models are based on significant variables inside the non-significant models. Additionally, some form of post-hoc analysis was performed such as Bonferroni correction should be considered.

Response. If the outcome variables can take on more than two values, the logistic regression model can be used. One of the most frequently performed diagnostic tests to check the goodness of fit of these models, the properties of which have been extensively studied is the Hosmer Lemeshow test (Hosmer & Lemeshow, 1980; Hosmer et al, 1998; Hosmer & Hjort, 2002). This test indicates model significance if the p-value measure is greater than 0.05, i.e., that there is no difference between the observed and predicted model. If this condition is fulfilled, the significant data within the model can be interpreted. In our case, we first checked the adequacy of the models by using as predictors the types of positive physical contact and as outcomes the levels of self-assessed depressive symptoms for both cohabiting (i.e., partners, children, relatives, or friends/roommates) and not cohabiting (i.e., partners, children, relatives and friends) people. The Bonferroni correction was proposed to circumvent the problem that as the number of tests increases, so does the likelihood of a type I error, i.e., concluding that a significant difference is present when it is not. However, the test has been subject to numerous criticisms, perhaps the most notable of which is that the likelihood of a type I error cannot decrease without increasing that of a type II error, so that real differences may not be detected (Perneger, 1998; Rothman, 1990; Ottenbacher, 1998; Moyé, 1998).

Source: https://doi.org/10.1080/03610928008827941

https://doi.org/10.1002/(SICI)1097-0258(19970515)16:9%3C965::AID-SIM509%3E3.0.CO;2-O

              https://doi.org/10.1002/sim.1200

              https://doi.org/10.1136/bmj.316.7139.1236

              https://www.jstor.org/stable/20065622

              https://doi.org/10.1093/oxfordjournals.aje.a009501

              https://doi.org/10.1016/S1047-2797(98)00003-9

Comment 4. The tables are filled with non-essential data and therefore very confusing. They should be merged and concentrated with only the most relevant data. Describing all sorts of statistical data could be described as a supplement. 

Response. We thank the reviewer for the valuable suggestion on how to improve the quality of our paper. Tracked changes have been made in the manuscript and non-essential material moved to the supplementary file.

Reviewer 2 Report

1. The tools you used do not measure "levels of depression" but they measure "levels of self-assessed depressive symptoms" at the most. Diagnosing depression requires evaluation by a physician, psychiatrist preferably. This error was found in abstract and in discussion.

2. I suggest a small elaboration in the methodology about how you used convienience sampling.

3. Please add approval number from the ethical committee.

4. Last but not least, the manuscript requires English editing. It is understandable, however a native needs to check it for minor linguistic errors.

I have no other comments.

Author Response

Comment 1 - The tools you used do not measure "levels of depression" but they measure "levels of self-assessed depressive symptoms" at the most. Diagnosing depression requires evaluation by a physician, psychiatrist preferably. This error was found in abstract and in discussion.

Response. Thank you for this comment. We have now replaced "levels of" with symptoms of anxiety, stress and depression throughout the manuscript (marked in red).

Comment 2 - I suggest a small elaboration in the methodology about how you used convenience sampling.

Response. We thank the reviewer for the advice on how to improve the methodological part of our manuscript. The requested changes can be read in lines 92 - 101.

Comment 3 - Please add approval number from the ethical committee.

Response. The approval number from the ethical committee was already reported in the "Institutional Review Board Statement" section. However, we have now added it in Participants and Procedure as well section (marked in red).

Comment 4 - Last but not least, the manuscript requires English editing. It is understandable, however a native needs to check it for minor linguistic errors.

Response. Thank you for this comment. One author of the manuscript is a native speaker of English and had edited the manuscript prior to submission. she has now made an even more careful reading and corrected possible grammar errors together with another native english researcher.